# Unsupervised Learning From Incomplete Measurements for Inverse Problems

**Julián Tachella**
Laboratoire de Physique
CNRS & ENSL
Lyon, France
julian.tachella@cnrs.fr

**Dongdong Chen**
School of Engineering
University of Edinburgh
Edinburgh, UK
d.chen@ed.ac.uk

**Mike Davies**
School of Engineering
University of Edinburgh
Edinburgh, UK
mike.davies@ed.ac.uk

## Abstract

In many real-world inverse problems, only incomplete measurement data are available for training which can pose a problem for learning a reconstruction function. Indeed, unsupervised learning using a fixed incomplete measurement process is impossible in general, as there is no information in the nullspace of the measurement operator. This limitation can be overcome by using measurements from multiple operators. While this idea has been successfully applied in various applications, a precise characterization of the conditions for learning is still lacking. In this paper, we fill this gap by presenting necessary and sufficient conditions for learning the underlying signal model needed for reconstruction which indicate the interplay between the number of distinct measurement operators, the number of measurements per operator, the dimension of the model and the dimension of the signals. Furthermore, we propose a novel and conceptually simple unsupervised learning loss which only requires access to incomplete measurement data and achieves a performance on par with supervised learning when the sufficient condition is verified. We validate our theoretical bounds and demonstrate the advantages of the proposed unsupervised loss compared to previous methods via a series of experiments on various imaging inverse problems, such as accelerated magnetic resonance imaging, compressed sensing and image inpainting.

## 1 Introduction

In multiple sensing applications, we observe measurements $y \in \mathbb{R}^m$ associated with a signal $x \in \mathcal{X} \subset \mathbb{R}^n$, through the forward process

$$y = Ax + \epsilon \tag{1}$$

where $A \in \mathbb{R}^{n \times n}$ is a linear measurement operator and $\epsilon$ denotes the noise affecting the measurements. This is the case of computed tomography [1], depth ranging [2] and non-line-of-sight imaging [3] to name a few. Estimating $x$ from $y$ is generally an ill-posed inverse problem due to the incomplete operator $A$ with $m < n$ and the presence of noise. Knowledge of the signal model is required to make this problem well-posed.

In many cases, obtaining ground-truth reconstructions $x$ to learn the reconstruction function $y \mapsto x$ might be very expensive or even impossible. For example, in medical imaging, it is not always possible to obtain fully sampled images of patients as they require long acquisition times. In astronomical imaging, it is impossible to obtain ground-truth references due to physical limitations. In electron-microscopy imaging [4], we can only measure 2D projections of a molecule. In these settings, we can only access measurements $y$ for learning. Moreover, if the measurement process $A$ is incomplete, it is fundamentally impossible to learn the model with only measurements $y$, as there

36th Conference on Neural Information Processing Systems (NeurIPS 2022).

is no information about the model in the nullspace of $A$. Thus, we end up with a chicken-and-egg problem: in order to reconstruct $x$ we need the reconstruction function, but to learn this function we require some reconstructed samples $x$.

This fundamental limitation can be overcome by using information from multiple incomplete sensing operators $A_1, \ldots, A_G$, the general principle being that each operator can provide additional information about the signal model if it has a different nullspace. For example, in the image inpainting problem, Studer and Baraniuk [5] used the fact that the set of missing pixels may vary between observed images to learn a sparse dictionary model and reconstruct the images. Yang et al. [6] used multiple operators to learn a Gaussian mixture model in the context of hyperspectral imaging and high-speed video. Bora et al. [7] exploited this idea for learning a generative model in various imaging problems such as deblurring and compressed sensing. Matrix completion methods [8] exploit a similar principle, as the missing entries of each column (i.e., signal) are generally different. Ideally we would like to learn the reconstruction function and signal model from only a small number of different measurement operators. We are thus motivated to determine typically how many such operators are required.

The problem can be formalized as follows. We first focus on the noiseless case to study the intrinsic identifiability problems associated to having only incomplete measurement data. The effect of noise will be discussed in Section 4. We assume that we observe a set of $N$ training samples $y_i$, where the $i$th signal is observed via $A_{g_i} \in \mathbb{R}^{m \times n}$, one of $G$ linear operators , i.e.,

$$y_i = A_{g_i} x_i \tag{2}$$

where $g_i \in \{1, \ldots, G\}$ and $i = 1, \ldots, N$. While we assume that the measurement operator $A_{g_i}$ is known for all observed signals, it is important to note that we do not know a priori if two observations $(y_i, A_{g_i})$ and $(y_{i'}, A_{g_{i'}})$ are related to the same signal as in Noise2Noise [9]. There are two natural questions regarding this learning problem:

**Q1. Signal Recovery** Is there a unique signal $x \in \mathcal{X}$ which verifies the measurements $y = A_g x$? In other words, is the reconstruction function $f : (y, A_g) \mapsto x$ one-to-one?

**Q2. Model Identification** Can we uniquely identify the signal model from measurement data alone obtained via the incomplete operators $A_1, \ldots, A_G$?

In general, there can be a unique solution for neither problem, just one or both. There might be a unique solution for signal recovery if the model is known, but it might be impossible to learn the model in the first place (e.g., blind compressed sensing [10]). The converse is also possible, that is, uniquely identifying a model without having enough measurements per sample to uniquely identify the associated signal (e.g., subspace learning from rank-1 projections [11]).

The answer to **Q1** is well-known from generalized compressed sensing theory, see for example [12]. Unique signal recovery is possible if the signal model is low-dimensional, i.e., if $A_g$ has $m > 2k$ measurements, where $k$ is the model dimension. On the other hand, **Q2** has been mostly studied in the context of matrix completion, where the set of signals is assumed to lie in a low-dimensional subspace of $\mathbb{R}^n$. Bora et al. [7] presented some results in the general setting, but only for the case where $G = \infty$ which is quite restrictive. In this paper, we provide sharp necessary and sufficient conditions which hold for any low-dimensional distribution (beyond linear subspaces) and only require a finite number of operators $G$.

If the conditions for signal recovery and model identification are satisfied, we can expect to learn the reconstruction function from measurement data alone. We introduce a new unsupervised learning objective which can be used to learn the reconstruction function $f : (y, A_g) \mapsto x$, and provides performances on par with supervised learning when the sufficient conditions are met. The main contributions of this paper are as follows:

- We show that unsupervised learning from a finite number of incomplete measurement operators is only possible if the model is low-dimensional. More precisely, we show that $m \geq n/G$ measurements per operator are necessary for learning, and that for almost every set of $G$ operators, $m > k + n/G$ measurements per operator are sufficient.

- We propose a new unsupervised loss for learning the reconstruction function that only requires incomplete measurement data, which empirically obtains a performance on par with fully supervised methods when the sufficient condition $m > k + n/G$ is met.

- A series of experiments demonstrate that our bounds accurately characterize the performance of unsupervised approaches on synthetic and real datasets, and that the proposed unsupervised approach outperforms previous methods in various inverse problems.

## 1.1 Related Work

**Blind Compressed Sensing**  The fundamental limitation of failing to learn a signal model from incomplete (compressed) measurements data goes back to blind compressed sensing [10] for the specific case of models exploiting sparsity on an orthogonal dictionary. In order to learn the dictionary from incomplete data, [10] imposed additional constraints on the dictionary, while some subsequent papers [13, 14] removed these assumptions by proposing to use multiple operators $A_g$ as studied here. This paper can be seen as a generalization of such results to more general signal models.

**Matrix Completion**  Matrix completion consists of inferring missing entries of a data matrix $Y = [y_1, \ldots, y_N]$, whose columns are generally inpainted samples from a low-dimensional distribution, i.e., $y_i = A_{g_i} x_i$ where the operators $A_{g_i}$ randomly select a subset of $m$ entries of the signal $x_i$. This problem can be viewed as the combination of model identification, i.e., identifying the low-rank subspace that the columns of $X = [x_1, \ldots, x_N]$ belong to, and signal recovery, i.e., reconstructing the individual columns. Assuming that the samples belong to a $k$-dimensional subspace can be imposed by recovering a rank-$k$ signal matrix $X$ from $Y$. If the columns are sampled via $G$ sufficiently different patterns $A_{g_i}$ with the same number of entries $m$, a sufficient condition [15] for uniquely recovering almost every subspace model is[1] $m \geq (1 - 1/G)k + n/G$.

A similar condition was shown in [16] for the case of *high-rank* matrix completion [17], which arises when the samples $x_i$ belong to a union of $k$-dimensional subspaces. We show that model identification is possible for almost every set of $G$ operators with $m > k + n/G$ measurements, however the theory presented here goes beyond linear subspaces, being also valid for general low-dimensional models.

**Deep Nets for Inverse Problems**  Despite providing very competitive results, most deep learning based solvers require measurements and signal pairs $(x_i, y_i)$ (or at least clean signals $x_i$) in order to learn the reconstruction function $y \mapsto x$ from incomplete measurements. A first step to overcome this limitation is due to Noise2Noise [9], where the authors show that it is possible to learn from only noisy samples. However, their ideas only apply to denoising settings where there is a trivial nullspace, as the operator $A$ is the identity matrix. This approach was extended in [18] to the case where two measurements are observed per signal, each associated with a different operator. Yaman et al. [19] and Artifact2Artifact [20] empirically showed that it is possible to exploit different measurement operators to learn the reconstruction function in the context of magnetic resonance imaging (MRI). AmbientGAN [7] proposed to learn a signal distribution from only incomplete measurements using multiple forward operators, however they only provide reconstruction guarantees for the case where an infinite number of operators $A_g$ is available[2], a condition that is not met in practice.

Another line of work focuses on learning using measurements from a single incomplete operator. The works in [21, 22] use the large system limit properties of random compressed sensing operators to learn from measurements alone. The equivariant imaging approach [23, 24] leverages invariance of the signal set to a group of transformations to learn from general incomplete operators.

## 2 Signal Recovery Preliminaries

We denote the nullspace of $A$ as $\mathcal{N}_A$. Its complement, the range space of the pseudo-inverse $A^\dagger$, is denoted as $\mathcal{R}_A$, where $\mathcal{R}_A \oplus \mathcal{N}_A = \mathbb{R}^n$ and $\oplus$ denotes the direct sum. Throughout the paper, we assume that the signals are sampled from a measure $\mu$ supported on the signal set $\mathcal{X} \subset \mathbb{R}^n$. Signal recovery has a unique solution if and only if the forward operator $x \mapsto y$ is one-to-one, i.e., if for every pair of signals $x_1, x_2 \in \mathcal{X}$ where $x_1 \neq x_2$ we have that

$$Ax_1 \neq Ax_2 \tag{3}$$

$$A(x_1 - x_2) \neq 0 \tag{4}$$

---

[1] A larger number of measurements $m = \mathcal{O}(k \log n)$ is required to guarantee a stable recovery when the number of patterns $G$ is large [8].

[2] Their result relies on the Cramér-Wold theorem, which is discussed in Section 3.

In other words, there is no vector $x_1 - x_2 \neq 0$ in the nullspace of $A$. It is well-known that this is only possible if the signal set $\mathcal{X}$ is low-dimensional. There are multiple ways to define the notion of dimensionality of a set in $\mathbb{R}^n$. In this paper, we focus on the upper box-counting dimension which is defined for a compact subset $S \subset \mathbb{R}^n$ as

$$\text{boxdim}(S) = \limsup_{\epsilon \to 0} \frac{\log N(S, \epsilon)}{-\log \epsilon} \tag{5}$$

where $N(S, \epsilon)$ is the minimum number of closed balls of radius $\epsilon$ with respect to the norm $\|\cdot\|$ that are required to cover $S$. This definition of dimension covers both well-behaved models such as compact manifolds and more general low-dimensional sets. The mapping $x \mapsto y$ is one-to-one for almost every forward operator $A \in \mathbb{R}^{m \times n}$ if [25]

$$m > \text{boxdim}(\Delta \mathcal{X}) \tag{6}$$

where $\Delta \mathcal{X}$ denotes the normalized secant set which is defined as

$$\Delta \mathcal{X} = \{\Delta x \in \mathbb{R}^n | \ \Delta x = \frac{x_2 - x_1}{\|x_2 - x_1\|}, x_1, x_2 \in \mathcal{X}, \ x_2 \neq x_1\}. \tag{7}$$

The term *almost every* means that the complement has Lebesgue measure 0 in the space of linear measurement operators $\mathbb{R}^{m \times n}$. The normalized secant set of models of dimension $k$ generally has dimension $2k$, requiring $m > 2k$ measurements to ensure signal recovery. For example, the union of $k$-dimensional subspaces requires at least $2k$ measurements[3] to guarantee one-to-oneness [26]. This includes well-known models such as $k$-sparse models (e.g., convolutional sparse coding [27]) and co-sparse models (e.g., total variation [28]). In the regime $k < m \leq 2k$, the subset of signals where one-to-oneness fails is at most $(2k - m)$-dimensional [25].

## 3  Uniqueness of Any Model?

A natural first question when considering uniqueness of the model is: can we recover any probability measure $\mu$ observed via forward operators $A_1, \ldots, A_G$, even in the case where its support $\mathcal{X}$ is the full $\mathbb{R}^n$? We show that, in general, the answer is no.

Uniqueness can be analysed from the point of view of the characteristic function of $\mu$, defined as $\varphi(w) = \mathbb{E}\{e^{iw^\top x}\}$ where the expectation is taken with respect to $\mu$ and $i = \sqrt{-1}$ is the imaginary unit. If two distributions have the same characteristic function, then they are necessarily the same almost everywhere. Each forward operator provides information about a subspace of the characteristic function as

$$\mathbb{E}\{e^{iw^\top A_g^\dagger y}\} = \mathbb{E}\{e^{iw^\top A_g^\dagger A_g x}\} \tag{8}$$

$$= \mathbb{E}\{e^{i(A_g^\dagger A_g w)^\top x}\} \tag{9}$$

$$= \varphi(A_g^\dagger A_g w) \tag{10}$$

where $A_g^\dagger A_g$ is a linear projection onto the subspace $\mathcal{R}_{A_g}$. Given that $m < n$, the characteristic function is only observed in the subspaces $\mathcal{R}_{A_g}$ for all $g \in \{1, \ldots, G\}$. For any finite number of operators, the union of these subspaces does not cover the whole $\mathbb{R}^n$, and hence there is loss of information, i.e., the signal model cannot be uniquely identified.

In the case of an infinite number of operators $G = \infty$, the Cramér-Wold theorem guarantees uniqueness of the signal distribution if all possible one dimensional projections ($m = 1$) are available [29, 7]. However, in most practical settings we can only access a finite number of operators and many distributions will be non-identifiable.

## 4  Uniqueness of Low-Dimensional Models

Most models appearing in signal processing and machine learning are assumed to be approximately low-dimensional, with a dimension $k$ which is much lower than the ambient dimension $n$. As discussed in Section 2, the low-dimensional property is the key to obtain stable reconstructions, e.g., in compressed sensing. In the rest of the paper, we impose the following assumptions on the model:

---

[3]While the bound in (6) guarantees *unique* signal recovery, more measurements (e.g., an additional factor of $\mathcal{O}(\log n)$ measurements) are typically necessary in order to have a *stable* inverse $f : y \mapsto x$, i.e., possessing a certain Lipschitz constant. A detailed discussion can be found for example in [12].

**A1** The signal set $\mathcal{X}$ is either

    (a) A bounded set with box-counting dimension $k$.

    (b) An unbounded conic set whose intersection with the unit sphere has box-counting dimension $k-1$.

This assumption has been widely adopted in the inverse problems literature, as it is a necessary assumption to guarantee signal recovery. Our definition of dimension covers most models used in practice, such as simple subspace models, union of subspaces (convolutional sparse coding models, $k$-sparse models), low-rank matrices and compact manifolds. It is worth noting that dimension is a property of the dataset and thus independent of the specific algorithm used for learning.

In the rest of the paper, we focus on conditions for the identification of the support $\mathcal{X}$ instead of the signal distribution $\mu$, due to the following observation: if there is a one-to-one reconstruction function (which happens for almost every $A$ with $m > 2k$ as explained in Section 2), uniqueness of the support implies uniqueness of $\mu$. If $\mathcal{X}$ is known and there is a measurable one-to-one mapping from each observed measurement $y$ to $\mathcal{X}$, then it is possible to obtain $\mu$ as the push-forward of the measurement distribution.

Before delving into the main theorem, we present a simple example which provides intuition of how a low-dimensional model can be learned via multiple projections $A_g$:

**Learning a one-dimensional subspace**    Consider a toy signal model with support $\mathcal{X} \subset \mathbb{R}^3$ which consists of a one-dimensional linear subspace spanned by $\phi = [1,1,1]^\top$, and $G = 3$ measurement operators $A_1, A_2, A_3 \in \mathbb{R}^{2\times3}$ which project the signals into the $x(3) = 0$, $x(2) = 0$ and $x(1) = 0$ planes respectively, where $x(i)$ denotes the $i$th entry of the vector $x$. The example is illustrated in Figure 1. The first operator $A_1$ imposes a constraint on $\mathcal{X}$, that is, every $x \in \mathcal{X}$ should verify $x(1) - x(2) = 0$. Without more operators providing additional information about $\mathcal{X}$, this constraint yields a plane containing $\mathcal{X}$, and there are infinitely many one-dimensional models that would fit the training data perfectly. However, the additional operator $A_2$ adds the constraint $x(2) - x(3) = 0$, which is sufficient to uniquely identify $\mathcal{X}$ as

$$\hat{\mathcal{X}} = \mathcal{X} = \{v \in \mathbb{R}^3 | \, v(1) - v(2) = v(2) - v(3) = 0\}$$

is the desired 1-dimensional subspace. Finally, note that in this case the operator $A_3$ does not restrict the signal set further, as the constraint $x(1) - x(3) = 0$ is verified by the other two constraints.

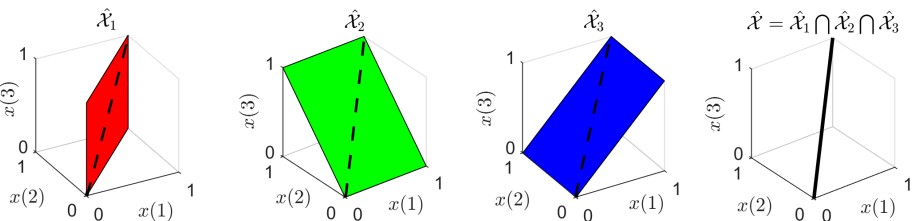

Figure 1: Toy example of a 1-dimensional subspace embedded in $\mathbb{R}^3$. If we only observe the projection of the signal set into the plane $x(3) = 0$, then there are infinite possible lines that are consistent with the measurements (red plane). Adding the projection into the $x(1) = 0$ plane, allows us to uniquely identify the signal model.

The ideas from the one-dimensional subspace example can be generalized and formalized as follows: for each projection $A_g$, we can constrain the model support $\mathcal{X}$ by considering the set

$$\hat{\mathcal{X}}_g = \{v \in \mathbb{R}^n | \, v = \hat{x}_g + u, \ \hat{x}_g \in \mathcal{X}, \ u \in \mathcal{N}_{A_g}\} \tag{11}$$

which has dimension at most $n - (m - k)$. Note that the true signal model is a subset of $\hat{\mathcal{X}}_g$. The inferred signal set belongs to the intersection of these sets

$$\hat{\mathcal{X}} = \bigcap_{g \in \mathcal{G}} \hat{\mathcal{X}}_g \tag{12}$$

which can be expressed concisely as

$$\hat{\mathcal{X}} = \{v \in \mathbb{R}^n | \, A_g(x_g - v) = 0, \ g = 1, \dots, G, x_1, \dots, x_G \in \mathcal{X}\} \tag{13}$$

Even though we have derived the set $\hat{\mathcal{X}}$ from a purely geometrical argument, the constraints in (13) also offer a simple algebraic intuition: the inferred signal set consists of the points $v \in \mathbb{R}^n$ which verify the following system of equations

$$\begin{bmatrix} A_1 \\ \vdots \\ A_G \end{bmatrix} v = \begin{bmatrix} A_1 x_1 \\ \vdots \\ A_G x_G \end{bmatrix}. \tag{14}$$

for all possible choices of $G$ points $x_1, \ldots, x_G$ in $\mathcal{X}$. In other words, given a dataset of $N$ incomplete measurements $\{A_{g_i} x_i\}_{i=1}^N$, it is possible to build $\hat{\mathcal{X}}$ by trying all the possible combinations of $G$ samples[4] and keeping only the points $v$ which are the solutions of (14).

It is trivial to see that $\mathcal{X} \subseteq \hat{\mathcal{X}}$, but when can we guarantee $\mathcal{X} = \hat{\mathcal{X}}$? As in the previous toy example, if there are not enough constraints, e.g., if we have a single $A$ and no additional measurement operators, the inferred set will have a dimension larger than $k$, containing undesired aliases. In particular, we have the following lower bound on the minimum number of measurements:

**Proposition 4.1** (Theorem 1 in [23])**.** *A necessary condition for model uniqueness from the measurement sets $\{A_g \mathcal{X}\}_{g=1}^G$ is that*

$$rank(\begin{bmatrix} A_1 \\ \vdots \\ A_G \end{bmatrix}) = n \tag{15}$$

*and thus $m \geq n/G$.*

*Proof.* In order to have model uniqueness, the system in (14) should only admit a solution if $v = x_1 = \cdots = x_G$. If the rank condition in (15) is not satisfied, there is more than one solution for any choice of $x_1, \ldots, x_G \in \mathcal{X}$. $\qquad\square$

Note that this necessary condition does not take into account the dimension of the model. As discussed in Section 3, a sufficient condition for model uniqueness must depend on the dimension of the signal set $k$. Our main theorem shows that $k$ additional measurements per operator are sufficient for model identification:

**Theorem 4.2.** *For almost every set of $G$ mappings $A_1, \ldots, A_G \in \mathbb{R}^{m \times n}$, under assumption **A1** the signal model $\mathcal{X}$ can be uniquely recovered from the measurement sets $\{A_g \mathcal{X}\}_{g=1}^G$ if the number of measurements per operator verifies $m > k + n/G$.*

The proof is included in the supplementary material. If we have a large number of independent operators $G \geq n$, Theorem 4.2 states that only $m > k + 1$ measurements are sufficient for model identification, which is slightly smaller (if the model is not trivial, i.e., $k > 1$) than the number of measurements typically needed for signal recovery $m > 2k$. In this case, it is possible to uniquely identify the model, without necessarily having a unique reconstruction of each observed signal. However, as discussed in Section 2, for $k < m \leq 2k$, the subset of signals which cannot be uniquely recovered is at most $(2k - m)$-dimensional.

**Operators with Different Number of Measurements**   The results of the previous subsections can be easily extended to the setting where each measurement operator has a different number of measurements, i.e. $A_1 \in \mathbb{R}^{m_1 \times n}, \ldots, A_G \in \mathbb{R}^{m_G \times n}$. In this case, the necessary condition in Proposition 4.1 is $\sum_{g=1}^G m_g \geq n$, and the sufficient condition in Theorem 4.2 is $\frac{1}{G} \sum_{g=1}^G m_g > k + n/G$. As the proofs mirror the ones of Proposition 4.1 and Theorem 4.2, we leave the details to the reader.

**Noisy measurement data**   Surprisingly, the results of this section are also theoretically valid if the measurements are corrupted by independent additive noise $\epsilon$, i.e., $y = A_g x + \epsilon$, as long as the noise distribution is *known* and has a nowhere zero characteristic function (e.g., Gaussian noise):

---

[4]Despite providing a good intuition, this procedure for estimating $\mathcal{X}$ is far from being practical as it would require an infinite number of observed samples if the dimension of the signal set is not trivial $k > 0$.

**Proposition 4.3.** *For a fixed noise distribution, if its characteristic function is nowhere zero, then there is a one-to-one mapping between the space of clean measurement distributions and noise measurement distributions.*

The proof is included in the supplementary material. If the clean measurement distribution can be uniquely identified from the noisy distribution, the results in Theorem 4.2 and Proposition 4.1 also carry over to the noisy setting. Note that this only guarantees model identifiability and makes no claims on the sample complexity of any learning process.

# 5 Algorithms

Unsupervised algorithms mainly come in two flavours: we can first learn a model $\hat{\mathcal{X}}$ to then reconstruct by projecting measurements into this set, or we can attempt to directly learn the reconstruction function parameterized by a deep network.

## 5.1 Learn the Model and Reconstruct

**Dictionary and Subspace Learning** If $\mathcal{X}$ is (approximately) a union of subspaces [10] or a single subspace [8], we can learn a model by

$$\arg\min_{z,D} \mathbb{E}_{(y,g)}\|y - A_g Dz\|^2 + \rho_1(D) + \rho_2(z) \tag{16}$$

where $\rho_1(D)$ and $\rho_2(z)$ are regularisation terms that promote low-dimensional solutions, e.g., sparse codes $z$ if $\mathcal{X}$ is a union of subspaces. At test time, the dictionary is fixed and the optimization is performed over the codes only.

**AmbientGAN** Complex datasets are often better modelled by a generative network $f : \mathbb{R}^k \mapsto \mathbb{R}^n$ whose input is a low-dimensional latent code $z \in \mathbb{R}^k$. The generative model can be learned using an adversarial strategy, i.e.,

$$\arg\min_f \max_d \mathbb{E}_{(y,g)} q\{d(A_g^\dagger y)\} + \mathbb{E}_z \mathbb{E}_g q\left\{1 - d\left(A_g^\dagger A_g f(z)\right)\right\} \tag{17}$$

where $d : \mathbb{R}^n \mapsto \mathbb{R}^n$ is the discriminator network which compares measurements in the image domain, $z$ is usually sampled from a Gaussian distribution, and $q(t) = \log(t)$ for standard GANs and $q(t) = t$ for Wasserstein GANs. At test time, the reconstruction can be obtained by finding the latent code that best fits the measurements $\hat{x} = f(\arg\min_z \|y - A_g f(z)\|^2)$, as in [30].

## 5.2 Learn to Reconstruct

Another approach consists in learning directly the reconstruction function $f : \mathbb{R}^m \times \mathbb{R}^{m \times n} \mapsto \mathbb{R}^n$ whose inputs are the measurement $y$ and the associated operator $A_g$, and the output is the reconstructed signal $x$. The reconstruction function can have either a denoiser form, $f(y, A_g) = \tilde{f}(A_g^\dagger y)$ where $\tilde{f}$ is independent of $A_g$ [1], or a more complex unrolled structure with many denoising and gradient steps [31].

**Measurement Splitting** Inspired by the Noise2Noise approaches [9, 18], some self-supervised methods [19, 20] split each measurement into two parts, $y^\top = [y_1^\top, y_2^\top]$, such that the input is $y_1$ and the target is $y_2$. These methods can be summarised as minimising the following loss

$$\arg\min_f \mathbb{E}_{(y,g)}\|y_2 - A_{g,2} f(y_1, A_{g,1})\|^2 \tag{18}$$

where $y_1 = A_{g,1}x + \epsilon_1$ and $y_2 = A_{g,2}x + \epsilon_2$. This approach suffers from the fact that $f$ does not use all the available information in a given measurement, as it attempts to solve a harder reconstruction problem associated with $y_1 = A_{g,1}x + \epsilon_1$. As reconstruction networks often fail to generalise to operators with more measurements [32], this method can suffer from suboptimal reconstructions at test time.

**Proposed Method**  The analysis in Section 4 shows that model identifiability necessarily requires that reconstructed signals are consistent with all operators $A_1, \ldots, A_G$. Thus, we propose an unsupervised loss that ensures consistency across all projections $A_g$, that is

$$\arg \min_f \mathbb{E}_{(y,g)} \left\{ \|y - A_g f(y, A_g)\|^2 + \mathbb{E}_s \|\hat{x} - f(A_s \hat{x}, A_s)\|^2 \right\} \tag{19}$$

where $\hat{x} = f(y, A_g)$. The first term ensures measurement consistency $y = A_g f(y, A_g)$, whereas the second term enforces consistency across operators, i.e., $f(y, A_g) = f(A_s f(y, A_g), A_s)$ for all $g \neq s$. Crucially, the second term prevents the network from learning the trivial pseudo-inverse $f(y, A_g) = A_g^\dagger y$. In practice, we choose an operator $A_s$ uniformly at random per minibatch. Thus, compared the supervised case, we only require an additional evaluation of $f$ and $A_s$ per minibatch. We coin this approach multi-operator imaging (MOI). This loss overcomes the main disadvantages of previous approaches: it doesn't require training a discriminator network, and it learns to reconstruct using all the available information in each measurement $y$.

## 6  Experiments

In this section, we present a series of experiments where the goal is to learn the reconstruction function with deep networks using real datasets. Experiments on low-dimensional subspace learning using synthetic datasets are presented in the supplementary material. All our experiments were performed using an internal cluster of 4 NVIDIA RTX 3090 GPUs with a total compute time of approximately 48 hs.

**Compressed Sensing and Inpainting with MNIST**  We evaluate the theoretical bounds on the MNIST dataset, based on the well known approximation of its box-counting dimension $k \approx 12$ [33]. The dataset contains $N = 60000$ training samples, and these are partitioned such that $N/G$ different samples are observed via each operator. The forward operators are compressed sensing (CS) matrices with entries sampled from a Gaussian distribution with zero mean and variance $m^{-1}$. The test set consists of 10000 samples, which are also randomly divided into $G$ parts, one per operator. To evaluate the theoretical bounds, we attempt to minimize the impact of the inductive bias of the networks' architecture [34, 35] by using a network with 5 fully connected layers and relu non-linearities.

Figure 2a shows the average test peak-signal-to-noise ratio (PSNR) achieved by the model trained using the proposed MOI loss for $G = 1, 10, 20, 30, 40$ and $m = 50, 100, 200, 300, 400$. The results follow the bound presented in Section 4 (indicated by the red dashed line), as the network is only able to learn the reconstruction mapping when the sufficient condition $m > k + n/G$ is verified. In sensing regimes below this condition, the performance is similar to the pseudo-inverse $A_g^\dagger$.

We also evaluate the reconstruction for a different number $G$ of random inpainting masks and different rates $m$. The inpainting operators have a diagonal structure which has zero measure in $\mathbb{R}^{m \times n}$, however our sufficient condition still provides a reasonable lower bound on predicting the performance, as shown in Figure 2b. It is likely that due to the coherence between measurement operators and images (both operators and MNIST images are sparse), more measurements are required to obtain good reconstructions than in the CS case.

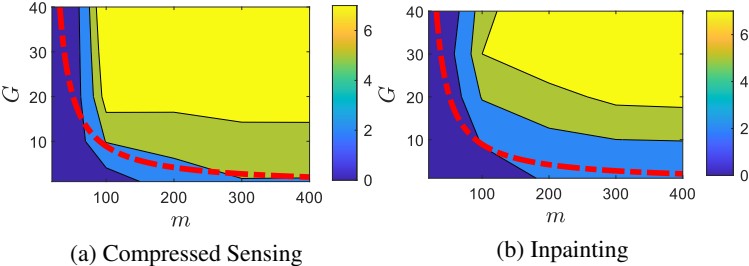

(a) Compressed Sensing        (b) Inpainting

Figure 2: Average test PSNR improvement in dB over the pseudo-inverse for the MNIST dataset using the proposed training loss, for different number of CS operators or inpainting masks and measurements per operator. The curve in red shows the necessary condition of Theorem 4.2, $m > k + n/G$.

| Inpainting/CelebA | $A^\dagger y$ | AmbientGAN | MOI (ours) | Supervised |
|---|---|---|---|---|
| | 9.05±1.65 | 29.57±1.24 | **34.05±3.77** | 36.21±3.76 |
| Acc. MRI/FastMRI | $A^\dagger y$ | Meas. Splitting | MOI (ours) | Supervised |
| Denoiser $f$ | 25.77±2.71 | 28.72±1.64 | **29.51±1.85** | 31.45±1.98 |
| Unrolled $f$ | 25.77±2.71 | 29.47±2.02 | **31.39±2.17** | 32.42±2.44 |

Table 1: Comparison of supervised and unsupervised learning methods for inpainting and accelerated MRI. Reported values correspond to average PSNR in dB on the testing set.

**Inpainting with CelebA**   We evaluate the unsupervised methods in Section 5 on the CelebA dataset [36], which is split into 32556 images for training and 32556 images for testing. We use the same U-Net (see the supplementary material for more details) for MOI and supervised learning, and the DCGAN architecture [37] for AmbientGAN as in [7]. We observed that using the U-Net for AmbientGAN's generator achieves worse results than the DCGAN architecture. Reconstructed test images are shown in Figure 3 and average test PSNR is presented in Table 1. The proposed method obtains an improvement of more than 4 dB with respect to AmbientGAN and falls only 2.1 dB behind supervised learning.

**Accelerated MRI with FastMRI**   Finally, we consider the FastMRI dataset [38], where the set of forward operators $A_g$ consist of different sets of single-coil $k$-space measurements, with $4\times$ acceleration, i.e., $m/n = 0.25$. We used 900 images for training and 74 for testing, which we split across $G = 40$ operators. We compare measurement splitting, MOI and supervised learning, all using the same denoiser or unrolled architecture (see the supplementary material for details). For measurement splitting, we follow the strategy in [19], and choose to assign a random subset representing 60% of the measurements in $A_g$ to $A_{g,1}$ and the remaining to $A_{g,2}$. We observed that a model trained with measurement splitting obtained less test error using reduced measurements associated with $A_{g,1}$ instead of the full measurements $A_g$, so we report the best[5] results using $A_{g,1}$. As observed in [32], the network fails to generalize to operators with more measurements. Average test PSNR is presented in Table 1. Reconstructed test images with the unrolled architecture are shown in Figure 4. The proposed method performs better than measurement splitting while obtaining results close to the supervised setting. All training approaches obtain a better performance with the unrolled architecture than using the denoising network due to the architectural improvements.

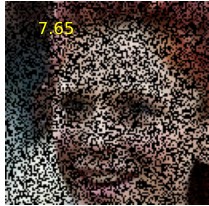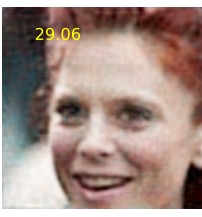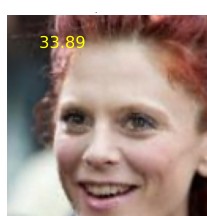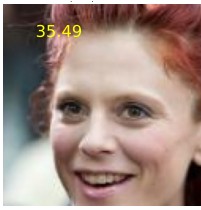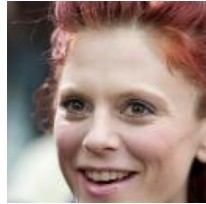

Figure 3: Reconstructed test images for the inpainting task using the CelebA dataset. From left to right: pseudo-inverse $A^\dagger y$, AmbientGAN, MOI, supervised and ground-truth. PSNR values are reported in yellow.

## 7   Limitations

Theorem 4.2 does not cover cases where the operators $A_g$ present some problem specific constraints (e.g., they are inpainting matrices) as well as cases where the signal model is only approximately low dimensional. Note however that Proposition 4.1 applies to constrained operators. We leave the study of sufficient conditions for these particular cases for future work. The proposed loss might not be effective in problems where learning the reconstruction function is impossible, e.g., due to very high noise affecting the measurements [4]. In this particular case, it might be possible to learn a generative model as in AmbientGAN [7].

---

[5]Using the full measurements with the denoiser architecture obtains an average test PSNR of 27.3 dB, i.e., a decrease of 1.4 dB with respect to the performance using $A_{g,1}$ presented in Table 1.

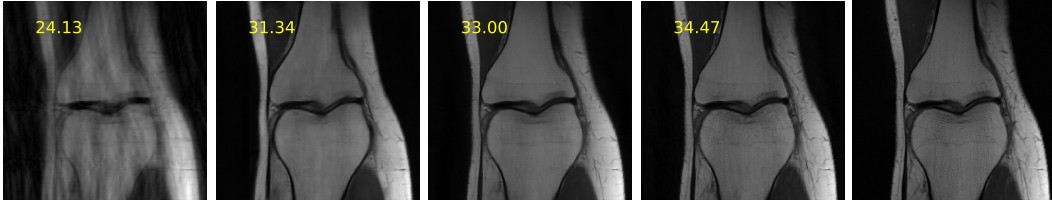

Figure 4: Examples of reconstructed test images for the accelerated MRI task using an unrolled network architecture (PGD-3). From left to right: pseudo-inverse $A^\dagger y$, measurement splitting, MOI, supervised and ground-truth. PSNR values are reported in yellow.

## 8    Conclusions

We presented sensing theorems for the unsupervised learning of signal models from incomplete measurements using multiple measurement operators. Our bounds characterize the interplay between the fundamental properties of the problem: the ambient dimension, the data dimension and the number of measurement operators. The bounds are agnostic of the learning algorithms and provide useful necessary and sufficient conditions for designing principled sensing strategies.

Furthermore, we presented a new practical unsupervised learning loss which learns to reconstruct incomplete measurement data from multiple operators, outperforming previously proposed unsupervised methods. The proposed strategy avoids the adversarial training in AmbientGAN, which can suffer from mode collapse [39], and, contrary to measurement splitting, is trained using full operators $A_g$. Our results shed light into the setting where access to ground truth data cannot be guaranteed which is of extreme importance in various applications.

## Acknowledgements

This work is supported by the ERC C-SENSE project (ERCADG-2015-694888).

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
