# Supplementary Material of Unsupervised Learning From Incomplete Measurements for Inverse Problems

**Julián Tachella**
Laboratoire de Physique
CNRS & ENSL
Lyon, France
julian.tachella@cnrs.fr

**Dongdong Chen**
School of Engineering
University of Edinburgh
Edinburgh, UK
d.chen@ed.ac.uk

**Mike Davies**
School of Engineering
University of Edinburgh
Edinburgh, UK
mike.davies@ed.ac.uk

## 1 Proofs

The proof of Theorem 4.2 in the main paper uses the following technical lemma:

**Lemma 1.1** (Lemmas 4.5 and 4.6 in [1]). *Let $S$ be a bounded subset of $\mathbb{R}^n$, and let $G_0, G_1, \ldots, G_t$ be Lipschitz maps from $S$ to $\mathbb{R}^m$. For each integer $r \geq 0$, let $S_r$ be the subset of $z \in S$ such that the rank of the $m \times t$ matrix*

$$\Phi_z = [G_1(z), \ldots, G_t(z)] \tag{1}$$

*is $r$, and let $boxdim(S_r) = k_r$. For each $\alpha \in \mathbb{R}^t$ define $G_\alpha(z) = G_0 + \Phi_z \alpha$. If for all integers $r \geq 0$ we have that $r > k_r$, then $G_\alpha^{-1}(0)$ is empty for almost every $\alpha \in \mathbb{R}^t$.*

*Proof.* The proof of Lemma 1.1 follows standard covering arguments and may be sketched as follows. From the dimensionality assumption, the set $S_r$ can be essentially covered by $\mathcal{O}(\epsilon^{-k_r})$ $\epsilon$-balls. Furthermore, for any $z \in S_r$, the probability (measured with respect to $\alpha \in \mathbb{R}^t$) that $G_\alpha(z)$ maps to the neighborhood of 0 scales as $\epsilon^r$. Hence the probability of this happening for any of the points in the cover scales as $\epsilon^{r-k_r}$. If we take $r > k_r$ then the probability of such an event tends to zero as we shrink $\epsilon$. Full details can be found in the proofs in [1]. $\square$

We can now present the proof of Theorem 4.2:

*Proof.* In order to have model uniqueness, we require that the inferred signal set $\hat{\mathcal{X}}$ defined as

$$\hat{\mathcal{X}} = \{v \in \mathbb{R}^n | \, A_g(x_g - v) = 0, \, g = 1, \ldots, G, x_1, \ldots, x_G \in \mathcal{X}\} \tag{2}$$

equals the true set $\mathcal{X}$, or equivalently that their difference

$$\hat{\mathcal{X}} \setminus \mathcal{X} = \{v \in \mathbb{R}^n \setminus \mathcal{X} | \, A_1(x_1 - v) = \cdots = A_G(x_G - v) = 0, \, x_1, \ldots, x_G \in \mathcal{X}\} \tag{3}$$

is empty, where $\setminus$ denotes set difference. Let $S \subset \mathbb{R}^{n(G+1)}$ be the set of all vectors $z = [v, x_1, \ldots, x_G]^\top$ with $v \in \mathbb{R}^n \setminus \mathcal{X}$ and $x_1, \ldots, x_G \in \mathcal{X}$. The difference set defined in (3) is empty if and only if for any $z \in S$ we have

$$\underbrace{\begin{bmatrix} -A_1 & A_1 & & \\ \vdots & & \ddots & \\ -A_G & & & A_G \end{bmatrix}}_{G_\alpha \in \mathbb{R}^{mG \times n(G+1)}} \underbrace{\begin{bmatrix} v \\ x_1 \\ \vdots \\ x_G \end{bmatrix}}_{z \in S} \neq 0 \tag{4}$$

$$G_\alpha(z) \neq 0 \tag{5}$$

36th Conference on Neural Information Processing Systems (NeurIPS 2022).

where $G_\alpha$ maps $z \in S$ to $\mathbb{R}^{mG}$. Let $\alpha = [\text{vec}(A_1)^\top, \ldots, \text{vec}(A_G)^\top]^\top \in \mathbb{R}^{mnG}$, then as a function of $\alpha$ we can also write (4) as

$$\begin{bmatrix} (x_1 - v)^\top \otimes I_m & & \\ & \ddots & \\ & & (x_G - v)^\top \otimes I_m \end{bmatrix} \alpha \neq 0 \tag{6}$$

where $\otimes$ is the Kronecker product and we used the fact that $A(x_g - v) = (x_g - v)^\top \otimes I_m \text{vec}(A)$. As $v$ does not belong to the signal set, the matrix on the left hand side of (6) has rank $mG$ for all $z \in S$. We treat the cases of bounded and conic signal sets separately, showing in both cases that, for almost every $\alpha \in \mathbb{R}^{mnG}$, the condition in (6) holds for all $z \in S$ if $m > k + n/G$:

**Bounded signal set** Let $S_\rho$ be a subset of $S$ defined as

$$S_\rho = \{z \in \mathbb{R}^{n(G+1)} \mid z = [v^\top, x_1^\top, \ldots, x_G^\top]^\top, x_1, \ldots, x_G \in \mathcal{X}, \|v\|_2 \leq \rho\}. \tag{7}$$

As $S_\rho$ is bounded, we have $\text{boxdim}(S_\rho) \leq kG + n$. Thus, if $mG > kG + n$, Lemma 1.1 states that for almost every $\alpha$, (6) holds for all $z \in S_\rho$. As $S$ can be decomposed as a countable union of $S_\rho$ of increasing radius, i.e., $S = \bigcup_{\rho \in \mathbb{N}} S_\rho$, and a countable union of events of measure zero has measure zero, then for almost every $\alpha$ all $z \in S$ verifies (6) if $m > k + n/G$.

**Conic signal set** If the signal set is conic, then $S$ is also conic. Hence, due to the linearity of (4) with respect to $z$, there exists $z \in S$ which does not verify (4) if and only if for any bounded set $B$ containing an open neighbourhood of 0, there exists a $z \in S \cap B$ which does not verify (4). As $\text{boxdim}(S \cap B) \leq Gk + n$, Lemma 1.1 states that for almost every $\alpha$, all $z \in S$ verifies (6) as long as $m > k + nG$.

$\square$

We end this section with the proof of Proposition 4.3 in the main paper:

*Proof.* Consider the noisy measurements associated to the $g$th operator $A_g$, as $z = y + \epsilon$, where $z$ are the observed noisy measurements, $y$ are the clean measurements and $\epsilon$ is additive noise (independent of $y$). The characteristic function of the sum of two independent random variables is given by the multiplication of their characteristic functions, i.e.,

$$\varphi_z(w) = \varphi_y(w)\varphi_\epsilon(w) \tag{8}$$

where $\varphi_z$, $\varphi_y$ and $\varphi_\epsilon$ are the characteristic functions the noisy measurement, clean measurements and noise distributions, respectively. If the characteristic function of the noise distribution is nowhere zero, we can uniquely identify the characteristic function of the clean measurement distribution as

$$\varphi_y(w) = \varphi_z(w)/\varphi_\epsilon(w) \tag{9}$$

The clean measurement distribution is fully characterized by its characteristic function $\varphi_y(w)$. We end the proof by noting that the same reasoning applies to the measurements of every operator $A_g$ with $g \in \{1, \ldots, G\}$. $\square$

## 2 Training Details

Algorithm 1 provides the pseudo-code of the proposed multi-operator imaging (MOI) method. The training details for each task are as follows:

**Compressed Sensing and Inpainting with MNIST.** In both cases, we use the Adam optimizer with a batch size of 128 and weight decay of $10^{-8}$. We use a fully connected network with 5 layers, where the number neurons in each layer are $784, 1000, 32, 1000, 784$ respectively. The nonlinearity is relu and the network has a residual connection between the input and output. For the CS task, we use an initial learning rate of $10^{-4}$ and train the networks for 1000 epochs, keeping the learning rate constant for the first 800 epochs and then shrinking it by a factor of 0.1. For the inpainting task, we use an initial learning rate of $5 \times 10^{-4}$ and train the networks for 500 epochs, keeping the learning rate constant for the first 300 epochs and then shrinking it by a factor of 0.1.

**Algorithm 1:** Pseudocode of MOI in a PyTorch-like style.

```
# 𝒢: forward operators 𝒢={1,…,G}
# f: reconstruction function (e.g., neural network)
for y, Aɡ in loader: # load a minibatch y with N samples and its corresponding operator Aɡ
    # randomly select a operator from 𝒢/g
    s = select(𝒢/g)
    x1 = f(y, Aɡ) # reconstruct x from y
    x2 = f(Aₛ(x1), Aₛ) # reconstruct x1
    # MOI training loss, Eqn.(18)
    loss = MSELoss(Aɡ(x1), y) # measurement consistency
         + MSELoss(x2, x1) # cross-operator consistency
    # update f network
    loss.backward()
    update(f.params)
```

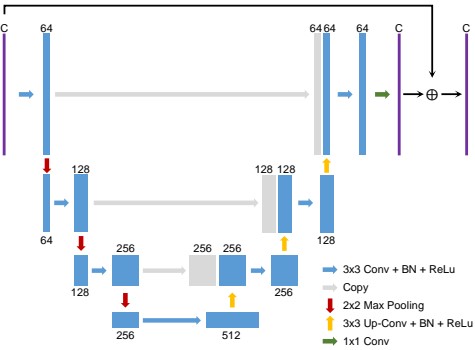

Figure 1: The residual U-Net used in the paper. The number of input and output channels is denoted as $C$, such that $C = 2$ for MRI and $C = 3$ for inpainting.

**Inpainting with CelebA.** The CelebA dataset contains more than 200K celebrity images, each with 40 binary attributes. We pick the attribute *smile* to evaluate the proposed method. The center part of the aligned images in the CelebA dataset are cropped to $128 \times 128$. We divide the selected images into two subsets for training and testing. There are 32557 images in each subset, which we split across $G = 40$ operators. For the inpainting task, we use the U-Net architecture (see Figure 1) to implement the reconstruction function $f(y, A)$, and the DCGAN architecture for AmbientGAN. Using the U-Net architecture for AmbientGAN's generator obtains an average test PSNR of $27.5 \pm 1.3$ dB, which is 2.1 dB below the performance obtained by the DCGAN generator reported in Section 6. We use Adam with a batch size of 20, an initial learning rate of $5 \times 10^{-4}$ and a weight decay of $10^{-8}$. We train the networks for 300 epochs, shrinking the learning rate by a factor of 0.1 after the first 200 epochs.

**Accelerated MRI with fastMRI.** For the denoiser network $f(y, A) = \tilde{f}(A^\dagger y)$, we use the U-Net in Figure 1 to implement $\tilde{f}$. For the unrolled network, we unfold the proximal gradient descent (PGD) algorithm (see Algorithm (10)) with $T = 3$ iterations. The step size is initialized as $\eta^{(t)} = 0.4$ and is then learned during training. We employ 3 U-Net networks using the architecture in Figure 1 to implement $f^{(t)}$ for $t = 1, 2, 3$ (no weight sharing across PGD iterations).

$$
\begin{array}{l}
\text{Unrolled Proximal Gradient Descent (PGD)} \\
\textbf{input: } y, A \\
x^{(0)} \leftarrow A^\dagger y \\
\text{for } t = 0, 1, \cdots, T-1 : \\
x^{(t+1)} \leftarrow f^{(t+1)}(x^{(t)} - \eta^{(t)} A^\top (A x^{(t)} - y)) \\
\text{end for} \\
\text{return } f(y, A) := x^{(T)}
\end{array}
\tag{10}
$$

We train the networks for 500 epochs with the Adam optimizer (batch size of 4), with initial learning rate $5 \times 10^{-4}$ for the first 300 epochs and then shrinking it by a factor of 0.1. In all experiments, we use complex-valued data and treat real and imaginary parts of the images as separate channels. For

the purpose of visualization, we display only the magnitude images. Reconstructed test images for the denoiser architecture are shown in Figure 2.

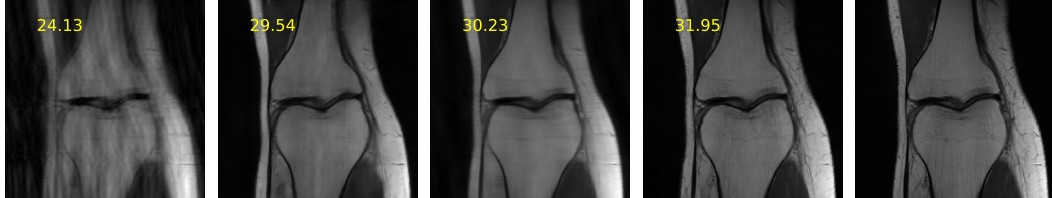

Figure 2: Examples of reconstructed test images for the accelerated MRI task using the fastMRI dataset using the denoiser architecture. From left to right: pseudo-inverse $A^\dagger y$, measurement splitting, MOI, supervised and ground-truth.

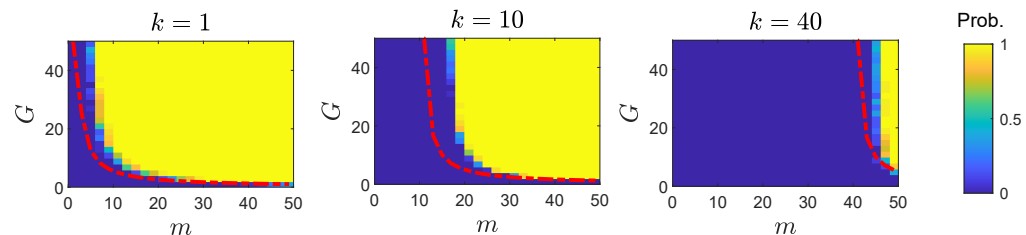

Figure 3: Reconstruction probability of a $k$-dimensional subspace using incomplete measurements arising from $G$ independent operators for different $k$. The curve in red shows the bound of Theorem 4.2, $m > k + n/G$.

# 3  Additional Experiments

## 3.1  Subspace Learning

We consider the problem of learning a $k$-dimensional subspace model from partial observations, where the signals $x_i$ are generated from a standard Gaussian distribution on the low-dimensional subspace. The observations $y_i$ are obtained by randomly choosing one out of $G$ operators $A_1, \ldots, A_G \in \mathbb{R}^{m \times n}$, each composed of iid Gaussian entries of mean 0 and variance $n^{-1}$. In order to recover the signal matrix $X = [x_1, \ldots, x_N]$, we solve the following low-rank matrix recovery problem

$$\arg\min_X \|X\|_*$$

(11)

$$\text{s.t. } A_{g_i} x_i = y_i \quad \forall i = 1, \ldots, N$$

where $\|\cdot\|_*$ denotes the nuclear norm. A recovery is considered successful if $\frac{\sum_i \|\hat{x}_i - x_i\|^2}{\sum_i \|x_i\|^2} < 10^{-1}$, where $\hat{x}_i$ is the estimated signal for the $i$th sample. We use a standard matrix completion algorithm [2] to solve (11). The ambient dimension is fixed at $n = 50$, and the experiment is repeated for $k = 1, 10, 40$. For each experiment we set $N = 150k$ in order to have enough samples to estimate the subspaces [3]. Figure 3 shows the probability of recovery over 25 Monte Carlo trials for different numbers of measurements $m$ and operators $G$. The reconstruction probability exhibits a sharp transition which follows the bound presented in Theorem 4.2, i.e., $m > k + n/G$.