# OpenReview forum: "Unsupervised Learning From Incomplete Measurements for Inverse Problems"
_NeurIPS.cc/2022/Conference — NeurIPS 2022 Accept_

### Official Review · Reviewer_YiZK · 2022-07-09

**Rating:** 3
**Confidence:** 5
**Soundness:** 1 poor
**Presentation:** 1 poor
**Contribution:** 1 poor

**Summary:**

This submission claims to characterize a sufficient condition ("model" being low dimensional) for the linear signal reconstruction problem to have the unique solution (?) in the unsupervised learning setting.

**Questions:**

- What is the setting of the unsupervised learning problem? There is no specification given in the submitted manuscript that clarifies this.

- Can authors elaborate what 'low-dimensional models' mean in this manuscript? This is unclear in theorem 4.2.

- In numerical experiment, line 288, what are the theoretical bounds authors are referring to?

**Strengths And Weaknesses:**

- The narrative is unclear as there is no clear statement on the setup of the main problem under consideration.
- It is likely that the authors mistake the term "model" with "signal".
- The statement of main results is vague.
- Numerical evidence is unconvincing.

---

> ### Author Response · Authors · 2022-08-02
> **Response to Reviewer YiZK**
>
> Thank you for your comments.
>
> > The narrative is unclear as there is no clear statement on the setup of the main problem under consideration.
>
> The problem setup is stated in lines (49-54) of the main paper, which reads:
>
> The problem can be formalized as follows.
> We first focus on the noiseless case to study the intrinsic identifiability problems associated to having only incomplete measurement data. The effect of noise will be discussed in Section 4.
> We assume that we observe a set of $N$ training samples $y_i$, where the $i$th signal is observed via $A_{g_i}\in \mathbb{R}^{m\times n}$, one of $G$ linear operators
> , i.e.,
> \begin{equation}
>     y_i = A_{g_i} x_i
> \end{equation}
> where $g_i \in \{1,\dots,G\}$ and $i=1,\dots,N$.  While we assume that the measurement operator $A_{g_i}$ is known for all observed signals, it is important to note that we do not know a priori if two observations $(y_i,A_{g_i})$ and $(y_{i'},A_{g_{i'}})$ are related to the same signal $x_i$.
>
> > It is likely that the authors mistake the term "model" with "signal".
>
> The difference between signal and model is stated in lines (57-60) of the main paper, and follows the convention used in compressed sensing theory. The model $\mathcal{X} \subset \mathbb{R}^{n}$ is a (low-dimensional) subset of the ambient space $\mathbb{R}^{n}$. For example, in compressed sensing $\mathcal{X}$ is often assumed to be the set of $k$-sparse vector, which is equivalent to a union of $k$-dimensional subspaces in $\mathbb{R}^{n}$. A signal is a point in the set $\mathcal{X}$, e.g., a $k$-sparse vector.
>
> > The statement of main results is vague.
>
> The main theoretical result is stated in Theorem 4.2. Please note that the signal set $\mathcal{X}$ is assumed to be low-dimensional model, according to the definition in lines 165-168. We can state this explicitly in the theorem.
>
> > Numerical evidence is unconvincing.
>
>  A range of experiments are set out in Section 6 and the Supplementary Materials which show that the empirical performance of deep networks and subspace learning algorithms are consistent with the theoretical results in Sections 3 and 4. Moreover, the experiments show the competitiveness of the proposed method with respect to other existing unsupervised approaches.
>
> > What is the setting of the unsupervised learning problem? There is no specification given in the submitted manuscript that clarifies this.
>
> Please see our answer above regarding the setup of the main problem.
>
> > Can authors elaborate what 'low-dimensional models' mean in this manuscript? This is unclear in Theorem 4.2.
>
> The definition of low-dimensional models is clearly stated in lines (165-168) of the main paper and uses the popular notion of box-counting dimension (Fractal geometry: mathematical foundations and applications, Falconer, 2004) as has been previously explored in the machine learning literature, e.g., (Intrinsic dimensionality estimation of submanifolds in $\mathbb{R}^{n}$, Hein and Audibert, 2005). In our paper, the formal definition reads:
>
>
> In the rest of the paper, we impose the following assumptions on the model:
>
> The signal set $\mathcal{X}$ is either
> *  A bounded set with box-counting dimension $k$.
> * An unbounded conic set whose  intersection with the unit sphere has box-counting dimension $k-1$.
>
> This assumption has been widely adopted in the inverse problems literature, as it is a necessary assumption to guarantee signal recovery. Our definition of dimension covers most  models used in practice, such as union of subspaces (simple subspace models, convolutional sparse coding models, $k$-sparse models), low-rank matrices and compact manifolds.
>
>
> > In numerical experiment, line 288, what are the theoretical bounds authors are referring to?
>
> We refer to the theoretical bounds associated with Theorem 4.2, under the assumption of box-counting dimension of the MNIST dataset is approximately 12, based on the well-known work [29] (Intrinsic dimensionality estimation of submanifolds in $\mathbb{R}^{n}$, Hein and Audibert, 2005). We will clarify this in the revised manuscript.
>
> If these answers have clarified some of your doubts regarding this paper, please consider raising your score.

---

### Official Review · Reviewer_AEvj · 2022-07-11

**Rating:** 7
**Confidence:** 4
**Soundness:** 3 good
**Presentation:** 3 good
**Contribution:** 3 good

**Summary:**

This paper studies unsupervised learning from multiple incomplete measurements for linear inverse problems. It formulates an interesting and sound problem where observed samples $y_i$ come from multiple measurement operators but are not necessarily related to the same signal $x$. Such a problem setting is practical and widely occurred in many real-world applications such as medical, biological and scientific imaging tasks. This paper first established a theoretical framework to identify the necessary and sufficient conditions for unsupervised signal recovery, which could be viewed as a generalized results for blind compressed sensing. Then it introduces an unsupervised loss for learning from incomplete measurements, inspired from the presented theoretical results. The experiments on compressed sensing and inpainting of natural images as well as accelerated MRI consistently demonstrate the superiority of the proposed method against baselines.


**Questions:**

[Theory part]
The problem formulation assumes signals are observed via G linear operators, but I don't find any claims of characterization of this set of operators. Are there any restrictions on these G operators, e.g., linear independency?

[Experiment part]
Could the authors provide a toy example in which the proposed unsupervised learning approach can compete with the supervised one, If the sufficient conditions in theory are meet. This would make the theory more convincing.

[Related work]
There are more related works studying unsupervised learning scheme for inverse problems [A,B,C]. Though the formulated problem settings are different, It would be very helpful to thoughly discuss the similarities and differences. Particularly, the loss function in [A] is very similar to the proposed one. Conducting ablation study with different unsupervised settings ([A] and this paper) and same reconstruction method (UNet) would be highly attractive.

[A] Training Image Estimators without Image Ground-Truth, NeurIPS 2019
[B] Unsupervised Learning with Stein’s Unbiased Risk Estimator, arxiv 2020
[C] Training deep learning based image denoisers from undersampled measurements without ground truth and without image prior, CVPR 2019

Overall it's a promising work from my perspective, and I would raise my score if the above issues could be addressed sufficiently.
***
**The rebuttal addresses my concerns, thus I raise my score to Accept**

**Strengths And Weaknesses:**

In my sense, this is a high-quality paper with both theoretical and empirical strengths. The studied problem setting is quite attractive and promising, which should be able to be applicable to a range of inverse problems in reality. The paper itself is nicely written and organized, even in the case fueled with mathematical details. I don't see severe weaknesses, though I do have questions/suggestion which might make the paper clearer and stronger [See Questions].

---

> ### Author Response · Authors · 2022-08-02
> **Response to Reviewer AEvj**
>
> Thank you for your comments.
>
> > [Theory part]
> The problem formulation assumes signals are observed via G linear operators, but I don't find any claims of characterization of this set of operators. Are there any restrictions on these G operators, e.g., linear independence?
>
> The necessary condition in Proposition 4.1 requires that the $G$ linear operators $A_1,\dots,A_G$ are (sufficiently) linearly independent, i.e., that the following stacked matrix has rank $n$:
> \begin{equation}
>      \text{rank} \left( \begin{bmatrix}
>     A_{1} \\\\
>     \vdots \\\\
>     A_{G}
>     \end{bmatrix} \right) = n
> \end{equation}
> which trivially results in the condition $m\geq n/G$.
>
> We will make this condition more explicit in the revised manuscript by writing Proposition 4.1 as "A necessary condition for model uniqueness is that \begin{equation}
>      \text{rank} \left( \begin{bmatrix}
>     A_{1} \\\\
>     \vdots \\\\
>     A_{G}
>     \end{bmatrix} \right) = n
> \end{equation} and thus $m\geq n/G$". We agree with the reviewer that this condition is more informative (and also useful for the practitioner) than simply $m\geq n/G$.
>
> > [Experiment part] Could the authors provide a toy example in which the proposed unsupervised learning approach can compete with the supervised one, if the sufficient conditions in theory are meet. This would make the theory more convincing.
>
> In Section 3.1 of the Supplementary Material, we perform experiments on the (toy) problem of subspace learning from synthetic data, where we can control the dimensionality of the subspace and use a convex inference algorithm which is assured to find the optimal solution. In this case, the unsupervised algorithm can obtain similar performances to the fully  supervised setting (i.e., accessing the whole data matrix) under some conditions on the number of samples observed. See also ("A Characterization of Deterministic Sampling Patterns for Low-Rank Matrix Completion", Pimentel-Alarcón et al., 2016) for a detailed study of the subspace learning setting.

---

> > ### Author Response · Authors · 2022-08-02
> > **Response to Reviewer AEvj (second part)**
> >
> > > [Related work] There are more related works studying unsupervised learning scheme for inverse problems [A,B,C]. Though the formulated problem settings are different, It would be very helpful to thoughly discuss the similarities and differences.
> > > [A] Training Image Estimators without Image Ground-Truth, NeurIPS 2019.
> > Particularly, the loss function in [A] is very similar to the proposed one.
> >
> > Thank you for highlighting this paper, we will add it to the references.
> > For the case of non-blind reconstruction (i.e., $A_g$'s are known) studied in our paper, the method in [A] takes a very similar form as that of (Yaman et al. 2018) and (Liu et al. 2020) cited in the paper.
> >
> > The algorithm in [A] requires 2 different measurements associated with the same signal $x$ and observed through different operators, i.e., $y_{1} = A_{g,1}x$ and $y_{2} = A_{g,2}x$, while we do not require this additional information. As explained in Section 5.2, it is however possible to split every measurement $y = A_{g}x$ into two smaller ones as in (Yaman et al., 2018), which is the approach described as "measurement splitting" in our paper.
> > The splitting method is compared directly with our proposed method in the accelerated MRI experiment in Section 6 where we obtain an improvement of .8 dB. Furthermore, as shown in Section 2 of the Supplementary Material, the splitting method fails to exploit all the available measurement information at test time, since it is trained to reconstruct from the partial operators $A_{g_1}$ and/or $A_{g_2}$ instead of the full $A_g$.
> >
> > > [B] Unsupervised Learning with Stein’s Unbiased Risk Estimator, arxiv 2020.
> >
> > > [C] Training deep learning based image denoisers from undersampled measurements without ground truth and without image prior, CVPR 2019
> >
> > Thank you for pointing out these papers, we will cite them in the revised manuscript. These papers tackle the problem of learning from a single operator $y=Ax$. As explained in Proposition 1 of (Equivariant Imaging: Learning Beyond the Range Space. Chen et al., 2021), learning from measurements of a single operator $A\in \mathbb{R}^{m\times n}$ is impossible if $m<n$ without imposing further properties on $A$, since the measurements contain no information in the nullspace of $A$. Our paper bypasses this limitation by using multiple operators, each with a potentially different nullspace.
> >
> > [B] considers either the case where $A=I$ (i.e. denoising) or the case where $A$ belongs certain class of random compressed sensing (CS) matrices (e.g., with iid Gaussian entries). In the former scenario the problem is not undersampled, and assuming knowledge of the noise characteristics, the SURE framework is used to learn a denoiser from the noisy signals. In the latter scenario, the CS matrices allow the back-projected measurements to be treated as noisy versions of the true signals and so the authors show that you can learn a denoiser within the Approximate Message Passing (AMP) framework combined with generalized SURE. [C] similarly focuses on the special case of compressed sensing matrices and learn a denoiser within the AMP+SURE framework.
> >
> > In fact, the SURE method is somewhat orthogonal/complementary to our approach, as it handles the noise present in the measurements, whereas the proposed method handles the fact that measurements are observed via incomplete operators with $m < n$. We believe that it should be possible to combine both approaches for unsupervised learning with noisy and incomplete measurements, i.e., using a SURE loss for the measurement consistency loss together with the loss proposed in our paper that ensures consistency across different operators. We leave a detailed analysis of the noisy setting for future work.

---

> > > ### Comment · Reviewer_AEvj · 2022-08-07
> > > **Thanks for the answers**
> > >
> > > Since the NeurIPS this year provides the opportunity to revise the paper in the rebuttal stage, it would be appreciated to directly take all the promised changes into a revision.

---

> > > > ### Author Response · Authors · 2022-08-08
> > > > **We have uploaded the revised manuscript**
> > > >
> > > > Thanks for the feedback. We have uploaded the revised manuscript, which includes the corrections mentioned in our response. We have also added the supplementary materials as an appendix in the main paper instead of a separated document to make the paper more self-contained.

---

### Official Review · Reviewer_GLHG · 2022-07-13

**Rating:** 8
**Confidence:** 4
**Soundness:** 4 excellent
**Presentation:** 4 excellent
**Contribution:** 4 excellent

**Summary:**

This paper presents a framework for unsupervised learning for inverse problems using multiple incomplete measurement models. The authors present theoretical results on number of measurements per model and number of models required for recovery assuming low-dimensional structure on the inputs. A training loss based on cycle consistency across different operators is suggested as a practical way to train these models. Experiments are performed to validate the theory and demonstrate on applications.

**Questions:**

Could you use the same network architecture for AmbientGAN as you use for MOI and supervised learning? This would aid the comparison.

It would also be instructive to experimentally see the impact of noise on recovery, as well as for different numbers of measurements per operator.


**Limitations:**

The authors describe the limitations of their work sufficiently.


**Strengths And Weaknesses:**

Strengths:
This paper builds off of prior work in low-dimensional signal recovery and very cleanly extends the results to fixed number of measurement operators and general low-dimensional structure. It also matches other known results like AmbientGAN for infinite measurement operators. In addition to developing a highly principled theory, the authors propose a novel and intutitive loss function that enforces a type of cycle consistency across operators. They show that this has a real effect on the result, compared to other schemes like SSDU that enforce cycle consistency across emeasurements. The paper is well written, the theory is well described, and the experimental results are strong. The


Weakness:
There appears to be a disconnect between the proposed training method and the theory. In other words, no analysis, intutition, or experimental verification is provided on the multi-operator learning. The authors do not discuss or show any type of convergence results for the training method.

The authors appear to use different network architectures for AmbientGAN and MOI in the Inpainting CelebA section. This makes direct comparison difficult.

---

> ### Author Response · Authors · 2022-08-02
> **Response to Reviewer GLHG**
>
> Thank you for your comments.
>
> > There appears to be a disconnect between the proposed training method and the theory. In other words, no analysis, intuition, or
> > experimental verification is provided on the multi-operator learning.
>
> The proposed cross-operator loss is inspired by our theoretical analysis which shows that model identifiability necessarily requires that reconstructed signals are consistent with all operators $A_1,\dots,A_{G}$, as illustrated in Figure 1 of the main paper. We will highlight this connection more clearly in the revised manuscript.
>
> > The authors do not discuss or show any type of convergence results for the training method.
>
> The theory presented in the paper provides necessary and sufficient conditions that are independent of the learning algorithm. For example, we show in the Supplementary Material that matrix completion algorithms follow the bound in Theorem 4.2. Our results shed light into the fundamental limitations of learning from incomplete data without any assumptions on the inference algorithm. Understanding the learning dynamics of deep networks is a challenging and interesting problem which is out of the scope of this paper, and constitutes an interesting avenue of future research.
>
> > Could you use the same network architecture for AmbientGAN as you use for MOI and supervised learning? This would aid the comparison.
>
> Unfortunately, we cannot use the same network for AmbientGAN and MOI, since AmbientGAN's generator maps a low dimensional noise input to a high dimensional image, whereas direct reconstruction methods such as MOI use networks whose input and output are of the same size.
> Reconstructing signals using ambientGAN also requires solving the non-convex optimization problem
> \begin{equation}
> \hat{x} = f({\arg\min}_z ||y - A_gf(z) ||^2)
> \end{equation}
> where $f$ is the generator, which might be harder if the latent space $z$ is higher dimensional.
> In addition, GAN-based models require an adversarial network which is not necessary in direct reconstruction methods such as MOI.
>
> Nonetheless, we tried using the U-Net architecture used in MOI for ambientGAN's generator (i.e., using a noise input of the same size as the output image), but the performance was significantly below the one obtained by the DCGAN generator used in (Bora et al., 2018). For the inpainting task with CelebA in Section 6, using the U-Net generator obtained an average test PSNR of 27.5$\pm$1.3 dB, which is 2.1 dB below the performance obtained by the DCGAN generator. We will include this result in the Supplementary Materials of the revised manuscript.
>
> Our theoretical results also work for AmbientGAN, i.e., they provide necessary and sufficient conditions for learning to be possible.
> The theory in Section 3 highlights the necessity of learning distributions with low dimensional support, and thus hint at the importance of the low dimensional noise input in AmbientGAN.
>
>
> > It would also be instructive to experimentally see the impact of noise on recovery, as well as for different numbers of measurements per operator.
>
> The current MOI approach is not designed to handle very noisy data, as the measurement consistency term might overfit the noise. However, if the noise distribution is known, we believe it is possible to extend MOI to handle noise by using a SURE-based measurement consistency loss (Unsupervised Learning with Stein’s Unbiased Risk Estimator, Metzler et al., 2020), (Robust Equivariant Imaging: a fully unsupervised framework for learning to image from noisy and partial measurements, Chen et al., 2022). We leave a detailed analysis of the noisy setting for future work.

---

> > ### Comment · Reviewer_GLHG · 2022-08-08
> > **Thank you for your response**
> >
> > Thank you for your response. I believe this is a strong paper and I will keep me current score. My one comment is that AmbientGAN can be applied in a similar manner as to your approach without the noise vector, I.e. according to the paper
> >
> > Fast unsupervised mri reconstruction without fully-sampled ground truth data using generative adversarial networks
> > Authors
> > Elizabeth K Cole, Frank Ong, Shreyas S Vasanawala, John M Pauly
> > Publication date
> > 2021
> > Conference
> > Proceedings of the IEEE/CVF International Conference on Computer Vision

---

### Meta-Review · Area_Chair_d896 · 2022-09-01

**Recommendation:** Accept
**Confidence:** Certain

**Metareview:**

This paper proposes an unsupervised learning algorithm for inverse problems using multiple incomplete measurement models. The authors presented theoretical results on the number of measurements per model and the number of models required for recovery under the assumption that the inputs have low-dimensional structures. In addition, a conceptually simple unsupervised learning loss is proposed and it only requires access to incomplete measurement data. The paper is well written, its theoretical analysis is strong, and the experimental results are convincing.

**Award:**

No

---

### Decision · Program_Chairs · 2022-09-14

Accept